# ❄️DueT🔥: Image-Text Contrastive Transfer Learning with Dual-adapter Tuning

**Taku Hasegawa**[1][*], **Kyosuke Nishida**[1][*], **Koki Maeda**[2][†], **Kuniko Saito**[1]
[1]NTT Human Informatics Laboratories    [2]Tokyo Institute of Technology
{taku.hasegawa, kyosuke.nishida}@ntt.com

## Abstract

This paper presents DueT, a novel transfer learning method for vision and language models built by contrastive learning. In DueT, adapters are inserted into the image and text encoders, which have been initialized using models pre-trained on uni-modal corpora and then frozen. By training only these adapters, DueT enables efficient learning with a reduced number of trainable parameters. Moreover, unlike traditional adapters, those in DueT are equipped with a gating mechanism, enabling effective transfer and connection of knowledge acquired from pre-trained uni-modal encoders while preventing catastrophic forgetting. We report that DueT outperformed simple fine-tuning, the conventional method fixing only the image encoder and training only the text encoder, and the LoRA-based adapter method in accuracy and parameter efficiency for 0-shot image and text retrieval in both English and Japanese domains.

## 1 Introduction

Pre-training of vision and language for learning the semantic correspondence between the two modalities has achieved impressive performance gains in a wide variety of downstream tasks. In particular, contrastive learning methods for image-text dual encoder models such as CLIP (Radford et al., 2021) and ALIGN (Jia et al., 2021) have attracted much attention. These models learn visual concepts directly from raw text about images by jointly training image and text encoders. A crucial aspect of the high performance of such models is that they are trained on a massive amount of image-text data. For example, CLIP was trained from scratch on a dataset of 400 million pairs collected from the Web without human annotations, and ALIGN was trained on a dataset of over one billion pairs. Thus, to build a model for a new non-English language

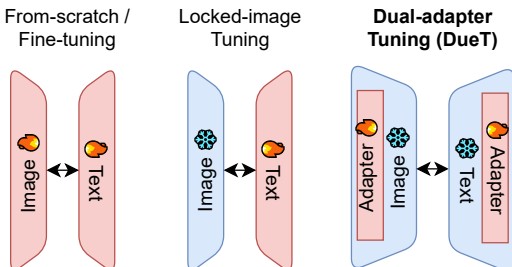

Figure 1: Design choices for transfer learning in image-text contrastive learning with dual encoders. *Left*: Learning all parameters from scratch or with fine-tuning. *Center*: LiT (Zhai et al., 2022) trains only the text encoder while keeping the pre-trained image encoder frozen. *Right*: Our proposal, **DueT** (dual-adapter tuning), trains only additional layers (adapters) inserted into each of the frozen pre-trained dual encoders.

or a domain not covered by the existing model, it necessitates collecting ample image-text pair data for the target language or domain prior to training. Another aspect is that the models are trained with substantial computational resources because it is important to form very large mini-batches for contrastive learning. Therefore, building on top of powerful pre-trained models with more data- and compute-efficient strategies is highly demanding. However, this issue has not yet been thoroughly investigated; most prior studies (Radford et al., 2021; Jia et al., 2021; Li et al., 2021; Singh et al., 2022) focused on training models from scratch using large-scale corpora. In an effort to address this challenge, LiT (Zhai et al., 2022), a parameter-efficient approach for developing vision and language models, has been proposed. It involves fixing the parameters of pre-trained uni-modal image encoders and exclusively training the text encoders. However, a comprehensive evaluation of the model's performance remains to be conducted.

In this study, we propose **DueT** (Dual-adapter Tuning), a method to learn the vision and language models. DueT uses a single-modal pre-trained

---
[*]Equal contribution.
[†]Contributed during internship at NTT Lab.

model as fixed values for the parameters of each encoder and adds adapters equipped with a gating mechanism to both encoders (Figure 1, right). In experiments in the English and Japanese domains, DueT outperformed simple transfer learning and LiT (Zhai et al., 2022). The main contributions of this study are as follows:

- **Transferability.** Our method transfers and connects knowledge acquired by pre-trained uni-modal encoders without catastrophic forgetting.

- **Parameter-efficiency.** It outperforms conventional methods of image-text pre-training while using fewer training parameters.

## 2 Related Work

### 2.1 Image-text Contrastive Learning

CLIP (Radford et al., 2021) and ALIGN (Jia et al., 2021) demonstrate that image-text dual encoder models pre-trained with contrastive objectives on a huge number of image-text pairs can learn strong image and text representations. They simply model the cross-modal interaction via the similarity of the global features of each modality and train the two encoders from scratch. In particular, there has been much research aimed at improving the cross-modal interaction and/or the pre-training strategy including objectives and transfer learning.

**Model architectures**. Dual encoder models have integrated cross-modal modules to fuse image and text features via cross-modal attention, with studies exploring cross-modal encoder (Li et al., 2021; Singh et al., 2022; Yuan et al., 2021), image-grounded text encoder-decoder (Li et al., 2022a), and cross-modal text decoder (Yu et al., 2022). Sharing layers between the image and text encoders has also been studied (You et al., 2022). Image-text contrastive learning has also been applied to cross encoders (Zhang et al., 2021) and a unified model was developed that can serve as either a dual encoder or a fusion encoder (Wang et al., 2021).

**Pre-training objectives**. Besides contrastive learning, pre-training objectives within each modality and across modalities have been also studied. For each modality, objectives inspired by uni-modal self-supervised pre-training schemes have been used for, e.g., masked language modeling (Li et al., 2021; Singh et al., 2022; Li et al., 2022b) and causal language modeling (Yu et al., 2022; Li et al., 2022a) masked image modeling (Singh

et al., 2022), image contrastive learning (Mu et al., 2021) and Siamese representation learning (Li et al., 2022b).

**Model initialization and freezing**. Most of the previous work on pre-training dual encoders falls into two parameter initialization categories: the from-scratch method, as in CLIP and ALIGN, randomizes all parameters in the image-text dual encoders (Yao et al., 2022; Yu et al., 2022; Yuan et al., 2021; Li et al., 2022b), while the fine-tuning method (Li et al., 2021; Singh et al., 2022; Li et al., 2022a) initiates the encoders from pre-trained uni-modal models. Recently, another line of research has appeared in the form of LiT (Zhai et al., 2022) and BASIC (Pham et al., 2021), which initialize the image encoder from a pre-trained model and fix its parameters during image-text contrastive learning, aligning closely with our study.

**Focus of this study**. This study focuses on the model architecture and transfer learning of a pair of uni-modal encoders. The key contribution is to incorporate trainable adapter layers into each frozen pre-trained uni-modal encoder. This represents a new line of research for pre-training image-text dual encoders. While this study does not focus on using cross-modal modules other than dual encoders and pre-training objectives other than contrastive learning, the recent advances described in this section can be incorporated into our model.

### 2.2 Adaptation of Pre-trained Models

**Parameter-efficient learning in NLP.**. Efficient adaptation methods of pre-trained language models for downstream tasks, such as adapter tuning (Houlsby et al., 2019; Pfeiffer et al., 2020, 2021; He et al., 2021; Mahabadi et al., 2021), prefix tuning (Li and Liang, 2021; Lester et al., 2021), additive methods (Guo et al., 2021; Hu et al., 2022; Zhang et al., 2020), and sparse-fine-tuning (Sung et al., 2021; Zaken et al., 2022), have been well-studied in the field of NLP. Unified frameworks combining these approaches have also been investigated (Mao et al., 2022; He et al., 2022).

**Adaptation of CLIP to downstream tasks.**. There have been several studies on extending the NLP approaches for adapting CLIP to various downstream vision-and-language tasks. To adapt CLIP encoders for image recognition tasks, CoOp (Zhou et al., 2022b) and CoCoOp (Zhou et al., 2022a) apply prefix tuning to the text encoder, while visual prompting (Bahng et al., 2022)

and VPT (Jia et al., 2022) to the image encoder. CLIP-Adapter (Gao et al., 2021) employs a feature adapter layer atop the image and the text encoder of CLIP for few-shot image recognition tasks. Tip-Adapter (Zhang et al., 2022) provides a training-free adaption for CLIP in order to conduct few-shot classification. VL-Adapter (Sung et al., 2022) and Flamingo (Alayrac et al., 2022) use the frozen image encoder of CLIP for visual features and perform adapter tuning on pre-trained language models. PAINT (Ilharco et al., 2022) uses interpolations between the model weights before and after fine-tuning on a task, without degrading accuracy on tasks where performance is already adequate.

**Focus of this study**. In contrast to prior studies adapting pre-trained CLIP encoders for downstream tasks, our focus was on enhancing the image-text contrast learning framework using robust pre-trained uni-modal encoders, avoiding catastrophic forgetting. We developed a model with data outside the language or domain of existing image-text models. We also examined the trade-off between performance and efficiency concerning additional parameters, and the model's performance relative to the number of training data. LilT (Khan and Fu, 2023), targeting efficient pre-training in parameter utilization, adopts a method similar to ours where adapters are inserted into the image and text encoders, which have been initialized using models pre-trained on uni-modal corpora and then frozen. The differences between LilT and our method are detailed in Section 3.2.

# 3 Methodology

The goal of this method is to efficiently construct vision and language models for diverse languages and domains, through knowledge transfer from uni-modal encoders such as ViT and BERT. To achieve this goal, we propose to train only gated adapter units (GAUs), which are designed for image-text contrastive learning, that are added to each of the frozen pre-trained dual encoders. This method can transfer and extend the excellent representations possessed by the pre-trained uni-modal encoders to the more challenging task of cross-modal understanding without requiring substantial fine-tuning of the original models. The GAUs learn to modulate and adapt the internal features of both the image and text encoders, effectively bridging the gap between the two modalities and enabling them to interact in a more meaningful way.

## 3.1 Overview

**Image-text contrastive learning framework.** A dual-encoder model that consists of an image encoder and a text encoder is trained to predict the correct image and text pairings. At run-time, each encoder maps the image and text to $d_m$-dimensional real-valued vectors, $x$ and $y$, respectively. The similarity between the input and the text is quantified by taking the dot product of their vectors, $s = x^\mathsf{T} y$.

**Encoder architecture.** DueT assumes that both the image and text encoders employ the Transformer architecture (Vaswani et al., 2017). DueT extends the Transformer block by inserting additional modules (Gated Adapter Units: GAUs) between layers, the details of which are described in Section 3.2. Here, we recapitulate the essentials of the Transformer architecture to aid readers in understanding the GAUs. Transformer encoders are composed of $L$ stacked blocks, where each block contains two types of sub-layer: multi-head self-attention and a fully connected feed-forward network (FFN). Each sub-layer has a residual connection and uses layer normalization (Ba et al., 2016). There are two approaches to applying layer normalization: the post-LN type such as BERT (Devlin et al., 2019), performs it after the sub-layer (outside the residual connection), while pre-LN type, such as ViT (Dosovitskiy et al., 2021), performs it before the sub-layer (inside the residual connection). Figure 2 shows a pre-LN type Transformer block.

**Input-output formulation.** Each encoder repeatedly transforms the input sequence of image patches or text tokens in $L$ blocks, resulting in a sequence of $d$-dimensional embeddings $\boldsymbol{H}^L = [\boldsymbol{h}_{\mathrm{CLS}}^L, \boldsymbol{h}_1^L, ..., \boldsymbol{h}_n^L, \boldsymbol{h}_{\mathrm{SEP}}^L] \in \mathbb{R}^{n \times d}$. The [CLS] and [SEP] tokens are special tokens inserted at the beginning and end of the input sequence. To calculate the similarity of the input and the text, we take the representation $\boldsymbol{h}_{\mathrm{CLS}} \in \mathbb{R}^d$ as the output of each encoder, and map it into the $d_m$-dimensional multi-modal embedding space $(x, y)$ with linear projections followed by $L_2$-normalization.

**Training objective.** We introduce a variant of unified contrastive learning (UniCL) (Yang et al., 2022). Given the $i$-th image-text pair, we generate a quadruple $(\boldsymbol{x}_i, \boldsymbol{y}_i, s_i, t_i)$ via encoders and hash algorithms, where $\boldsymbol{x}_i \in \mathbb{R}^{d_m}$ and $s_i$ are respectively the embeddings and the hash value of the image and $\boldsymbol{y}_i \in \mathbb{R}^{d_m}$ and $t_i$ are those of the text. We treat all image-text-pairs associated with the hash values

of the pair as positives $\mathcal{P}_i$ and all other random image-text pairs that can be formed in a training batch $\mathcal{B}$ as negatives. The loss to be minimized is defined as the sum of two losses:

$$\mathcal{L}_{\text{i2t}} = -\frac{1}{|\mathcal{B}|} \sum_{i \in \mathcal{B}} \frac{1}{|\mathcal{P}_i|} \sum_{k \in \mathcal{P}_i} \log \frac{\exp(\boldsymbol{x}_i^\mathsf{T} \boldsymbol{y}_k / \tau)}{\sum_{j \in \mathcal{B}} \exp(\boldsymbol{x}_i^\mathsf{T} \boldsymbol{y}_j / \tau)}, \quad (1)$$

$$\mathcal{L}_{\text{t2i}} = -\frac{1}{|\mathcal{B}|} \sum_{i \in \mathcal{B}} \frac{1}{|\mathcal{P}_i|} \sum_{k \in \mathcal{P}_i} \log \frac{\exp(\boldsymbol{y}_i^\mathsf{T} \boldsymbol{x}_k / \tau)}{\sum_{j \in \mathcal{B}} \exp(\boldsymbol{y}_i^\mathsf{T} \boldsymbol{x}_j / \tau)}, \quad (2)$$

where $k \in \mathcal{P}_i = \{k \mid k \in \mathcal{B}, s_i = s_k \text{ or } t_i = t_k\}$. $\tau$ is a trainable parameter of the temperature to scale the logits. It is different from Florence (Yuan et al., 2021), which uses UniCL, in that it utilizes the image hash value in addition to the text hash value for constructing the positive set. In this study, the MD5 hash algorithm was used.

## 3.2 Gated Adapter Units

Unlike previous adapters (Houlsby et al., 2019; Khan and Fu, 2023) that have only a feed-forward network (FFN), the GAUs also have a gating mechanism to facilitate transfer learning of the two pre-trained image and text encoders without catastrophic forgetting. Gate types can be categorized into two main classifications: adaptive, where gate values depend on the input to the adapter, and static, where they do not; moreover, they can be characterized as either element-wise, with gate values determined individually for each element, or layer-wise, where values are decided per layer. Preliminary experiments found negligible differences among these categories; consequently, for the sake of simplicity, we have chosen to adopt the static and layer-wise approach in this paper. Further details concerning the adaptive type can be found in Appendix G. The FFN, which is independently and identically applied to each position, consists of two linear transformations with a nonlinear activation $\phi$ in between them. The dimensionality of the input and output is $d$, and the inner-layer has dimensionality $m$.

$$\text{GAU}^l(\boldsymbol{H}^l) = \alpha^l \text{FFN}^l(\text{LN}(\boldsymbol{H}^l)) + (1 - \alpha^l) \boldsymbol{H}^l, \quad (3)$$

$$\text{FFN}^l(\boldsymbol{h}) = \phi(\boldsymbol{h} \boldsymbol{W}_{\text{down}}^l + \boldsymbol{b}_{\text{down}}^l) \boldsymbol{W}_{\text{up}}^l + \boldsymbol{b}_{\text{up}}^l, \quad (4)$$

where the input $\boldsymbol{H}^l$ is the output after the residual connection of the FFN module of the Transformer. LN denotes layer normalization (Ba et al., 2016). [1]

---

[1] When inserting a GAU into a pre-LN type Transformer(e.g. ViT (Dosovitskiy et al., 2021);fig2) that uses an LN within the residual connections of the sub-modules, insert the LN before the FFN, as shown in Equation (1). For the post-LN type (e.g. BERT (Devlin et al., 2019)), insert the LN after the FFN.

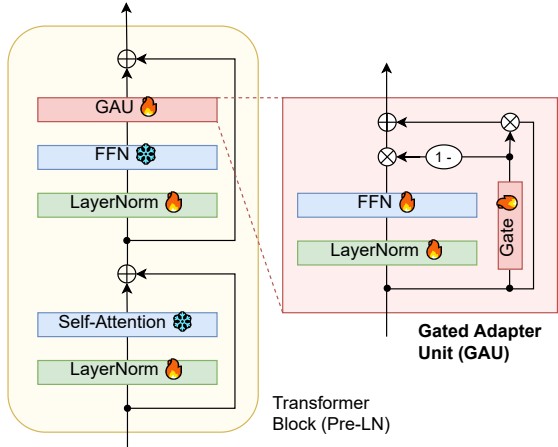

Figure 2: **Gated adapter unit** (**GAU**) and its integration with the Transformer block. *Left*: a GAU is inserted after the feed-forward network in each Transformer block. *Right*: a GAU is an FFN having a gating mechanism. If the gating mechanism outputs 0, GAU becomes the identity function.

$\boldsymbol{W}_{\text{down}} \in \mathbb{R}^{d \times m}$, $\boldsymbol{b}_{\text{down}} \in \mathbb{R}^m$, $\boldsymbol{W}_{\text{up}} \in \mathbb{R}^{m \times d}$, $\boldsymbol{b}_{\text{up}} \in \mathbb{R}^d$ and $\alpha^l \in \mathbb{R}$ are independent trainable parameters for each GAU. Let $d$ and $m$ denote the dimensions of the input-output and intermediate layers, respectively. We use the GeLU activation for $\phi$ and a sigmoid function for $\sigma$, respectively. With the gated residual connection, the GAU makes use of the gating mechanism to adaptively transform or bypass ($\alpha^l = 0$) the signals.

Unlike the previous study (Houlsby et al., 2019; Khan and Fu, 2023) in which adapter layers were added both after the self-attention and after the FFN sub-layer within each Transformer block, we place the GAU after the FFN sub-layer, as shown in Figure 2.

## 3.3 Design choices for dual-adapter tuning

**Parameter initialization and freezing.** The parameter initialization and freezing strategies for each encoder involve several design choices. We define four potential settings as an extension of the LiT definition (Zhai et al., 2022):

- **from-scratch** (u). Parameters are randomly initialized and trainable (*unlocked*).

- **fine-tuning** (U). Parameters are initialized from a pre-trained model and trainable (*Unlocked*).

- **locked-tuning** (L). Parameters are initialized from a pre-trained model and frozen (*Locked*).

- **GAU tuning** (g). The main parameters are initialized from a pre-trained model and frozen (except for layer normalization). The extra parameters of the *gated adapter units* (§3.2) are randomly initialized (except for the gating value $\alpha^l$) and trainable.

- **LoRA tuning** (l). The main parameters are initialized from a pre-trained model and frozen (except for layer normalization). The extra parameters of the *LoRA unit* (Hu et al., 2022) are randomly initialized and trainable.

The settings can be changed for each encoder. For example, Lu denotes locked-image tuning (LiT) (Zhai et al., 2022), which trains the text encoder from scratch while keeping the pre-trained image encoder (LiT-FS); LU denotes an extension of locked-image tuning, which keeps the pre-trained image encoder frozen and fine-tunes the text encoder (LiT-FT); gg denotes the setting of our dual-adapter tuning (DueT); uu denotes the from-scratch setting used in (Radford et al., 2021; Jia et al., 2022) (FS); UU denotes the standard fine-tuning setting used in (Li et al., 2021; Singh et al., 2022) (FT); and ll denotes the setting of the LoRA tuning (LoRA). Note that DueT updates all parameters in the layer normalization of the pre-trained encoders during training, as it was done in previous adapter-tuning studies on NLP (Houlsby et al., 2019; He et al., 2021). DueT initializes the gating value $\alpha^l$ to 0.02. Refer to Appendix H.1 for LoRA implementation specifics.

**Parameter efficiency.** Adapter units can reduce the number of additional parameters with a small bottleneck dimension $m$, with total parameters in a single adapter, inclusive of gating mechanism, as $(2d + 1)(m + 1)$. Given equivalent hidden size $d$ and layers $L$ for image and text encoders, total adapter parameters are approximately $8Ldm$. $m$ is a simple trade-off lever between performance and parameter efficiency. See Appendix A for further detail on model size and parameter efficiency.

# 4 Experiments

First, we compared DueT with the standard image-text contrastive learning frameworks. [2]

## 4.1 Pre-trained Models

We mainly used the architecture and pre-trained weights of **ViT-B/16-AR-IN21k** (Dosovitskiy et al., 2021) and **BERT-base** (Devlin et al., 2019) for the image and text encoders, if not stated otherwise. The details are given in Appendix B.

## 4.2 Datasets

### 4.2.1 Pre-Training Datasets

We used the following two pre-training datasets consisting of image and text pairs with different visual concepts and languages.

**YFCC-CLIP**. The YFCC100M (Thomee et al., 2016) contains the metadata on 99.2M images and 0.8M videos from Flickr (2004-2014). (Radford et al., 2021) defines a 15M image subset, termed YFCC-CLIP, filtered for English titles/descriptions [3]. It's split into 14,760,364, 10,000, and 10,000 pairs for training, development, and test sets. Note that the visual concepts of YFCC-CLIP are relatively close to those acquired by the image encoders pre-trained on ImageNet-21k, which contains a large number of photos collected from Flickr.

**JWeb Dataset**. We collected a private dataset of image and Japanese-text pairs form a variety of publicly available sources on the Web. We mainly used 5M, 10K, 10K pairs for the training(JWeb-5M), development, and test sets. Compared with YFCC-CLIP, the JWeb dataset has different visual concepts and contains various images, including shopping items, slides, PC screens, and Japanese characters. The details are given in Appendix C.

### 4.2.2 Evaluation Datasets

**Zero-shot image classification.** We evaluated zero-shot transfer of models in visual classification tasks on ImageNet ILSVRC-2012 benchmark (Deng et al., 2009) (IN). We also used the Japanese version: ImageNet-JP (IN-JP) (translated from ImageNet). We used the top-1 accuracy as the evaluation metric.

**Zero-shot image-text retrieval.** We also evaluated the zero-shot transfer performance of models in both text and image retrieval on the **MS-COCO** (Lin et al., 2014) and **Flickr30k** (Plummer et al., 2017) and the test set of the pre-training corpus (YFCC-CLIP or JWeb5M). We also used the Japanese versions of these datasets: **STAIR**

---

[2]While LilT (Khan and Fu, 2023) is a contemporaneous study and utilizes different training data, a direct comparison through experiments is not conducted in this paper. However, we provide a discussion based on the insights gained from our study in Appendix L.

[3]https://github.com/openai/CLIP/blob/main/data/yfcc100m.md

**Captions** (Yoshikawa et al., 2017) (translated from MS-COCO) and **Flickr30k-JP** (Nakayama et al., 2020) (translated from Flickr30k). The details are shown in Appendix D. As in (Li et al., 2021), the evaluation metric was the mean value of Recall@k (R@k; $k = 1, 5, 10$) defined as the fraction of queries for which the correct item is contained in the closest $k$ items to the query.

## 4.3 Results

Table 1 shows the results of the evaluation of the English and Japanese models. DueT outperformed the baseline evaluation in 0-shot image and text retrieval. In particular, it outperformed fine-tuning (from-scratch) on the Flickr30k-JP image retrieval by 4.2 (39.2) points, while reducing the number of training parameters to about 58%. While fine-tuning achieved the highest baseline score and excelled on the YFCC-CLIP test set, it underperformed in 0-shot retrievals due to over-fitting in English and Japanese training data. The performance deficit was particularly noticeable when the image encoder's pre-training and training set domains differed in the Japanese model evaluation. LiT-FS and LiT-FT underperformed DueT in the evaluation of both the Japanese and English models because of their poor adaptability to domains not covered by the pre-training of the image encoder. In the case of from-scratch, which does not perform transfer learning, we found that a training data sizes of 5-15M are insufficient. While the LoRA tuning scored lower than that of the proposed method, it is very parameter-efficient and achieved the same score as LiT-FS with a small number of training parameters. A comparison between the proposed method with a smaller number of training parameters and LoRA is shown in 4.4.

## 4.4 Ablation studies

We conducted evaluations on GAU, which is the main contribution of our research. In all of the following experiments, the JWeb5M dataset was used as the training data.

**Is it possible to achieve parameter-efficient learning?** The proposed method changes the number of training parameters depending on the dimension of the intermediate representation of the FFNs in the inserted GAU. We investigated how the number of training parameters affects the performance of the model. Table 2 shows that accuracy improves the number of training param-

eters increases. On the other hand, a small number of additional parameters did not cause a significant performance loss: compared with fine-tuning, which updated 210.0M parameters, DueT updated 15.1M (4.5M) parameters; its performance on STAIR (Flickr30k-JP) was equivalent to that of fine-tuning, and parameter-efficient learning was achieved. DueT with 2.7M parameter updates, which is almost the same number as that of LoRA (3.2M parameter updates), significantly outperformed LoRA on all of the evaluation sets. These results indicate that DueT is a parameter-efficient method.

**How effectively do the GAUs work at each layer?** As shown in Table 3, performance decreased when the number of Transformer blocks into which GAUs were inserted was reduced, especially when adapter insertion was limited to the image encoder only. This result shows that inserting adapters in both the image encoder and text encoder is important for image-text contrastive learning. On the other hand, removal of the adapters from either encoder, with the exception of the region near the output layer of the image encoder, resulted in a minimal performance deterioration. However, there was a notable decrease in performance when the adapters were removed from the vicinity of the image encoder's output layer and also when the total number of inserted adapters was substantially reduced.

**How do the gates perform at each layer?** As shown in Figure 3, the gate values of the image encoder tended to increase closer to the output layer. This indicates that the effect of the FFN in the GAU increases near the output layer, which further suggests that the knowledge of the pre-trained model used for the initial values is utilized around the input layer and new knowledge is acquired by the adapter is utilized around the output layer. On the other hand, the gate value for the text encoder was around 0.4 for all layers, which is not as extreme as the gate value for the image encoder. This result indicates that the text encoder uses the knowledge of the pre-trained model as the initial values to create the embedding of the image-text pairs in all layers. The detailed discussion of gate values is provided in Appendix I.

**Does the initial gate value and gate learning have an influence?** Table 4 contrasts cases with and without gate values (fixed $\alpha = 1.0$), demonstrating

| Method | #TP EN | #TP JP | EN (trained on YFCC-CLIP) YFCC-CLIP I→T | YFCC-CLIP T→I | MS-COCO† I→T | MS-COCO† T→I | Flickr30k† I→T | Flickr30k† T→I | IN† | JP (trained on JWeb5M) JWeb test I→T | JWeb test T→I | STAIR† I→T | STAIR† T→I | Flickr30k-JP† I→T | Flickr30k-JP† T→I | IN-JP† |
|---|---|---|---|---|---|---|---|---|---|---|---|---|---|---|---|---|
| FS | 195.5 | 210.0 | 78.88 | 78.47 | 43.26 | 27.2 | 64.5 | 42.9 | 25.29 | 50.62 | 50.83 | 33.31 | 23.54 | 40.77 | 30.01 | 14.9 |
| FT | 195.5 | 210.0 | 88.83 | 88.28 | 61.37 | 40.27 | 83.93 | 62.77 | 47.3 | 72.12 | 72.95 | 60.98 | 50.29 | 78.7 | 65.04 | 37.23 |
| LiT-FS | 109.7 | 124.2 | 67.39 | 66.6 | 52.92 | 35.47 | 75.97 | 55.71 | 45.18 | 50.7 | 48.78 | 48.59 | 32.92 | 66.5 | 51.23 | 35.78 |
| LiT-FT | 109.7 | 124.2 | 71.35 | 69.71 | 57.69 | 38.0 | 78.87 | 57.18 | 48.46 | 54.77 | 53.23 | 52.62 | 36.39 | 72.37 | 54.92 | 38.49 |
| LoRA$_8$ | 1.5 | 1.5 | 53.14 | 53.02 | 52.16 | 37.41 | 75.57 | 59.1 | 40.51 | 52.62 | 52.02 | 47.34 | 39.6 | 65.97 | 56.1 | 32.8 |
| DueT | 56.8 | 57.6 | 86.22 | 85.33 | **61.75** | **42.05** | **84.5** | **64.2** | **55.43** | 73.17 | 73.21 | 61.89 | 52.53 | 81.4 | 69.27 | **42.85** |

Table 1: Performance of models trained on YFCC-CLIP and JWeb5M. Text and Image Retrieval (I→T, T→I) Performance. We used BERT-base and ViT-B/16 and set $m = 1{,}536$ for DueT and $r = 8$ for LoRA. #TP indicates the number of trainable parameters in millions. IN(IN-JP) denotes ImageNet. † Zero-shot retrieval task.

| | $m$ | #TP | STAIR I→T | STAIR T→I | Flickr30k-JP I→T | Flickr30k-JP T→I | IN-JP |
|---|---|---|---|---|---|---|---|
| LoRA$_8$ | | 1.5 | 47.34 | 39.6 | 65.97 | 56.1 | 32.8 |
| LoRA$_{16}$ | | 3.2 | 47.71 | 39.95 | 65.93 | 55.7 | 33.64 |
| | 48 | 2.7 | 55.54 | 45.88 | 75.63 | 62.94 | 36.25 |
| | 96 | 4.5 | 57.33 | 48.0 | 78.1 | 65.72 | 39.62 |
| | 192 | 8.0 | 59.28 | 49.87 | 80.3 | 67.53 | 40.86 |
| | 384 | 15.1 | 60.41 | 50.48 | 80.43 | 69.33 | 43.1 |
| | 768 | 29.2 | 62.02 | 52.27 | **82.33** | 69.77 | 43.45 |
| | 1536 | 57.6 | 61.89 | **52.53** | 80.4 | 69.27 | 42.85 |
| | 3072 | 114.2 | **62.16** | 52.38 | 81.47 | **70.74** | **44.43** |
| FT | | 210.0 | 60.98 | 50.29 | 78.7 | 65.04 | 37.23 |

Table 2: Zero-shot transfer performance of models trained on JWeb5M with DueT and different inner dimensions of FFN in adapters. #TP indicates the number of trainable parameters in millions. We used the same value of $m$ for all adapters in both encoders. Light blue cells represent the same level of performance as fine-tuning (FT). Indices 8 and 16 in LoRA$_8$ and LoRA$_{16}$ represent $r$ in LoRA.

| Image | Text | STAIR I→T | STAIR T→I | Flickr30k-JP I→T | Flickr30k-JP T→I | Ave. |
|---|---|---|---|---|---|---|
| 1-12 | N/A | 52.63 | 45.23 | 69.9 | 63.01 | 57.69 |
| 1-12 | 8-12 | 60.78 | 51.11 | 79.33 | **69.66** | 65.22 |
| 1-12 | 4-12 | 61.43 | 52.02 | 81.8 | 69.35 | 66.15 |
| 1-12 | 1-4 | 59.71 | 50.98 | 80.0 | 68.71 | 64.85 |
| 1-12 | 1-8 | 60.79 | 52.17 | **81.87** | 68.79 | 65.91 |
| N/A | 1-12 | 58.83 | 47.4 | 75.73 | 62.43 | 61.10 |
| 8-12 | 1-12 | 60.12 | 50.46 | 77.83 | 65.47 | 63.47 |
| 4-12 | 1-12 | **61.89** | 52.01 | 79.6 | 69.09 | 65.65 |
| 1-4 | 1-12 | 59.0 | 48.97 | 77.57 | 65.88 | 62.86 |
| 1-8 | 1-12 | 61.29 | 50.71 | 79.23 | 67.61 | 64.71 |
| 8-12 | 8-12 | 57.95 | 49.4 | 76.7 | 66.83 | 62.72 |
| 4-12 | 4-12 | 61.45 | 51.37 | 80.6 | 69.59 | 65.75 |
| 1-4 | 1-4 | 56.69 | 47.13 | 75.4 | 63.49 | 60.68 |
| 1-8 | 1-8 | 60.03 | 50.61 | 79.97 | 68.78 | 64.85 |
| 1-12 | 1-12 | **61.89** | **52.53** | 80.4 | 69.27 | 66.02 |

Table 3: Performance of models trained on JWeb15M with adapters inserted in different ranges of layers. Ave. represents the average of STAR and Flickr30k-JP.

| $\alpha_{\text{init}}$ | fixed | JWeb test I→T | JWeb test T→I | STAIR I→T | STAIR T→I | Flickr30k-JP I→T | Flickr30k-JP T→I |
|---|---|---|---|---|---|---|---|
| 1.0 | ✓ | 73.18 | 72.65 | 60.94 | 50.86 | 77.7 | 68.08 |
| 1.0 | | 72.36 | 72.24 | 59.79 | 50.6 | 77.63 | 69.23 |
| 0.02 | ✓ | 68.43 | 67.83 | 59.43 | 50.91 | 78.7 | 67.53 |
| 0.02 | | **73.17** | **73.21** | **61.89** | **52.53** | **80.4** | **69.27** |

Table 4: Performance of models trained on JWeb5M with different initial gate values. We set $m = 1{,}536$.

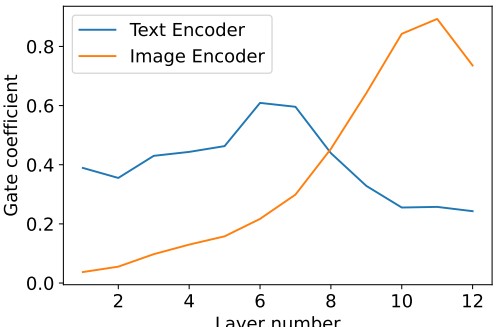

Figure 3: Gate values for each layer. Orange: represents the gate values of the image encoder. Blue: represents the gate values of the text encoder.

the efficacy of gate values introduction. Furthermore, performance was improved by updating the gate value $\alpha$ through learning. When the initial gate value was set large, the impact of unlearned GAUs was strong in the early stage of learning, making it difficult for the transfer learning to progress.

**How does the number of training data affect performance?** Figure 4 shows the performance of each method trained on various numbers of training data in the JWeb Dataset. The details in the dataset are shown in Appendix C. The proposed method outperformed fine-tuning or LoRA when the number of training data was between 0.5M and 10M. The score of fine-tuning approached that of DueT as the number of training data increased,

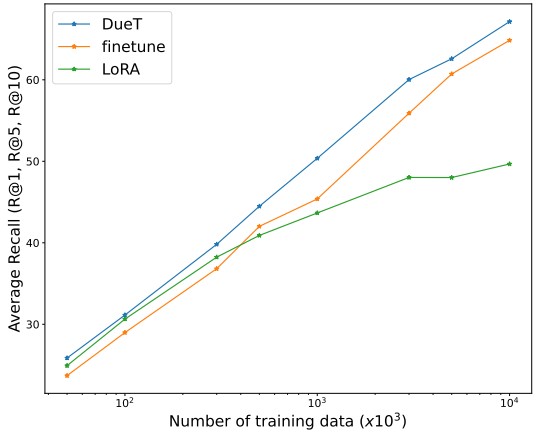

Figure 4: Performance of each model versus number of training data. **Horizontal**: Number of training data in the JWeb dataset (log). **Vertical**: average of 'recall(@1, @5, @10) of STAIR, Flickr30k-jp, and accuracy of ImageNet.

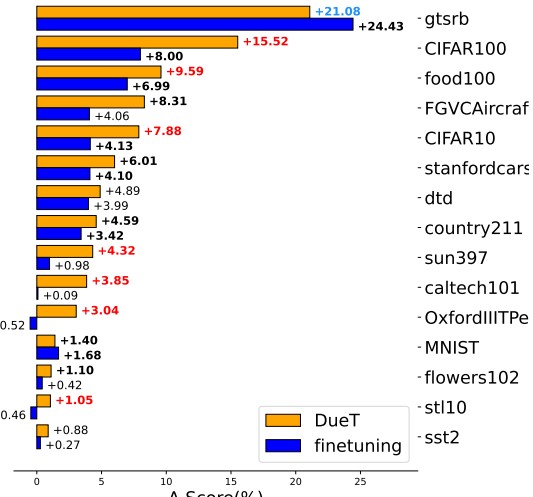

Figure 5: Linear Probe on uni-modal ViT vs. linear probe on FT-CLIP and DueT. **Bold**: Scores within the 99.5% Clopper-Pearson confidence interval showing significant difference from uni-modal ViT. Red: Scores where DueT significantly outperformed FT, determined by the 99.5% Clopper-Pearson confidence interval. Blue: Scores where DueT significantly underperformed FT, determined by the 99.5% Clopper-Pearson confidence interval.

while LoRA's score was relatively higher than that of fine-tuning when the number of training data was small. This result shows that DueT is effective even when the amount of training data is small.

### 4.5 Linear Probing

We investigated the effect of DueT and fine-tuning on the image encoders used for initialization using contrastive learning for text and image matching. On the basis of DueT and FT models trained on JWeb-5M in 4.3, linear probing was performed for the downstream task. The details of each task and experiment are presented in Appendix F.1 and F.2.

Figure 5 depicts the scores of DueT and fine-tuning relative to those of linear probing with ViT-B/16-AR-IN21k. The results in the figure show that the contrastive learning of DueT lead to its higher scores on many tasks compared with the uni-modal ViT. The results in the figure show that DueT had higher scores on almost all tasks (on 13 out of 15 tasks significantly) compared with the uni-modal ViT. Furthermore, compared with FT, DueT had a higher percentage increase in scores on seven tasks. With the exception of GTSRB, DueT performed at a level equivalent to FT in all other tasks. This suggests that DueT prevent catastrophic forgetting of knowledge and successfully combines adaptation to a new domain with knowledge utilization.

### 5 Conclusion

We propose DueT, which performs transfer learning by using adapter tuning (Houlsby et al., 2019) in the pre-training of CLIP (Radford et al., 2021), a visual and language model constructed by contrastive learning. A single-modal pre-trained model is used as fixed values for the parameters of each encoder, and adapter GAUs equipped with a gating mechanism extended from the one described in (Houlsby et al., 2019) are added to both encoders for training. Transfer learning of uni-modal pre-trained models is an important topic in regard to CLIP, which requires a large amount of training data. The fact that we were able to devise a method that is superior in performance and parameter efficiency to fine-tuning, LiT (Zhai et al., 2022), and LoRA (Hu et al., 2022) is a major contribution to research on vision and language models. The results of the experimental evaluations also gave us insight into the number of additional parameters, number of training data, and necessity of gates for transfer learning in the construction of Japanese CLIP. The results of this research will contribute to the development of services such as dialogue, navigation, and content generation and retrieval that require integrated visual and linguistic understanding.

## Limitations

This study explored only classification and retrieval as zero-shot transfer tasks. It leaves as future work the evaluation of zero-shot transfer for a broader set of tasks such as detection, segmentation, visual question answering, and image captioning. In particular, we have tested the effectiveness of DueT in the setting of learning image-text matching via contrastive learning, and it will be important to test its effectiveness on the above tasks with LM-based VLMs, such as (Alayrac et al., 2022) and (Yu et al., 2022).

This study demonstrates that DueT is a parameter-efficient method, particularly in instances of limited data size and computational resources. On the other hand, the fine-tuned model could outperform DueT when there was a large amount of data and sufficient computing resources available. While the results of this study suggest that DueT can save computational costs within a certain budget, it may be useful to consider the fine-tuned setup as well, given a sufficient budget.

DueT is designed to be adaptable to to non-English languages and domains outside the training domain of existing CLIP models. However, this study is limited to validating DueT in Japanese and English; it is not yet clear whether it will be effective in other languages. In addition, we leave for future work the analysis of the effects of differences in encoders that have been pre-trained on uni-modal data used to initialize learning.

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

## A Model size and parameter efficiency

As shown in section 3.3, the parameter efficiency of DueT can be adjusted by setting hyper-parameter $m$ to a moderate value. For example, when we used ViT-B/16 and BERT$_{base}$ ($L = 12$ and $d = 768$) and set $m$ to 96, the number of parameters updated by DueT was around 7 million, or about $3.64\%$ of the total number of parameters of the two original models (110 million plus 96 million).

Note, however, that the addition of GAUs also increases the total number of parameters in the model. Table 5 shows the relationship between the model size and the number of training parameters for each method.

## B Models

**Image Encoders.** We used **ViT-B/16-AR-IN21k**, which is a 12-layer Vision Transformer (Dosovitskiy et al., 2021) pre-trained on ImageNet-21k (Deng et al., 2009) (14 million images, 21,843 classes) with AugReg (Steiner et al., 2021) at resolution 224x224[4]. The number of hidden states $d$ was 768, and images were presented as a sequence of 16x16 patches.

**Text Encoders.** We used **BERT-base** as the text encoder, which is a 12-layer Transformer pre-trained on lower-cased English Wikipedia and BookCorpus (Devlin et al., 2019)[5]. We also newly constructed its a Japanese version pre-trained with lower-cased Japanese Wikipedia and CC-100-ja (Conneau et al., 2020). The number of hidden states $d$ was 768.

## C JWeb dataset

The JWeb dataset is the dataset we constructed for this study. It is based on images and Japanese captions that we collected from a wide range of web sites. Images were resized at the time of downloading to have a short side size of 256 or larger. A trained fastText (Joulin et al., 2017) model[6] was used to determine the language. The distribution of the number of words in the caption when word

---

[4]github.com/google-research/vision_transformer

[5]huggingface.co/bert-base-uncased

[6]https://fasttext.cc/docs/en/language-identification.html

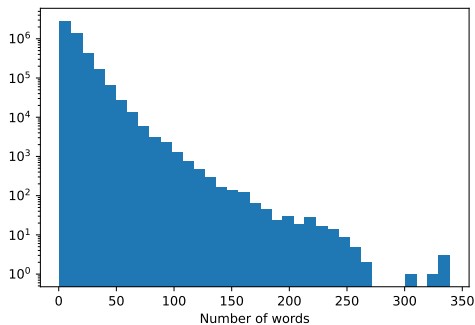

Figure 6: Distribution of the number of words for each caption in JWeb-5M. Horizontal axis: number of words (bin width 10), Vertical axis: frequency (log)

segmentation was performed by mecab-unidic is shown in Fig. 6. The mean (median) number of characters and words was 26.3 (20.0) characters and 11.9 (9.0) words. Captions consisting of one or two words were included instead of natural sentences. When 100 images were sampled for the study, 32 images contained Japanese characters (Kanji, Hiragana, or Katakana).

**JWeb-k** From the JWeb dataset, we constructed multiple training sets with different numbers of training data. The training sets are denoted as JWeb-k, where $k$ is the number of image-text pairs in the training data (e.g., JWeb-500k, JWeb-1m). Note that the training set with the smaller number of data was a subset of the larger training set, and each image-text pair was randomly selected. Table 6 shows the statistics of the training sets.

**JWeb-5M** Unless otherwise stated, we utilized JWeb-5M consisting of 5M image-text pairs for training the model. The number of unique images in this dataset is 4,942,737 (confirmed by the md5 hash value) and the number of captions is 4,369,144. The top-five most frequently used captions were "landscape", "woman," "food," "cat," and "cherry blossom.

## D Evaluation dataset

MS-COCO (Lin et al., 2014) contains 123K images, each accompanied with five manually written captions. We followed the data split used in (Karpathy and Fei-Fei, 2017) and used the 5K test set.

Flickr30K contains 31K images collected from the Flickr website, with five textual descriptions per image. We follow the data split used in (Karpathy and Fei-Fei, 2017) and use the 1K test set.

| Method | Not. | $m(r)$ | #Param. | #Train Param. |
|---|---|---|---|---|
| FS | uu | - | 210.0M (100%) | 210.0M (100%) |
| FT | UU | - | 210.0M (100%) | 210.0M (100%) |
| LiT-FS | Lu | - | 210.0M (100%) | 124.2M (59.1%) |
| LiT-FT | LU | - | 210.0M (100%) | 124.2M (59.1%) |
| $LoRA_8$ | aa | 8 | 210.6M (100.3%) | 1.5M (0.71%) |
| $LoRA_{16}$ | aa | 16 | 211.1M (100.5%) | 2.0M (0.95%) |
| $LoRA_{32}$ | aa | 32 | 212.3M (101.1%) | 3.2M (1.52%) |
| $LoRA_{64}$ | aa | 64 | 214.7M (102.2%) | 5.6M (2.67%) |
| $LoRA_{256}$ | aa | 256 | 228.9M (102.2%) | 19.7M (2.67%) |
| $LoRA_{1024}$ | aa | 1024 | 285.5M (102.2%) | 76.4M (2.67%) |
| DueT | aa | 48 | 211.8M (100.9%) | 2.7M (1.28%) |
| DueT | aa | 96 | 213.6M (101.7%) | 4.5M (2.3%) |
| DueT | aa | 192 | 217.3M (103.5%) | 8.0M (4.09%) |
| DueT | aa | 384 | 224.2M (106.8%) | 15.1M (7.72%) |
| DueT | aa | 768 | 238.4M (113.5%) | 29.2M (14.9%) |
| DueT | aa | 1536 | 266.7M (127.0%) | 57.6M (29.5%) |
| DueT | aa | 3072 | 323.4M (154.0%) | 114.2M (58.4%) |

Table 5: Number of parameters with ViT-B/16 ($d = 768$) and BERT-base ($d = 768$). #Param.: Number of parameters in the model at inference. #Train Param.: Number of trainable parameters.

| Training set | #Images | #Captions |
|---|---|---|
| JWeb-50K | 49,993 | 49,073 |
| JWeb-100K | 99,973 | 97,299 |
| JWeb-300K | 299,786 | 286,910 |
| JWeb-500K | 499,415 | 473,266 |
| JWeb-1M | 997,637 | 930,435 |
| JWeb-3M | 2,979,041 | 2,686,802 |
| JWeb-5M | 4,942,737 | 4,369,144 |
| JWeb-10M | 9,781,186 | 8,365,458 |

Table 6: Statistics of each training set. #Images: Number of unique images. #Captions: Number of unique captions.

The images were collected from Flickr, and natural text captions were created by a crowd-worker. nltk-punct's mean (median) word count was 11.3 (eleven) words. For the Japanese version of STAIR Captions (Yoshikawa et al., 2017), five new Japanese captions were created for each image by the crowd-worker for the same images as COCO. Flickr30K had 31.5 (31) words.

Flickr30K contains 31K images, each with five captions. The mean (median) number of words is 31.4 (12). The Japanese version, Flickr30k-JP (Nakayama et al., 2020), was created by expertly translating each Flickr30k caption. The mean (median) word count was 18.4 (17) words. For COCO and Flickr30k, the images in the test data were selected according to the partitioning of the dataset in COCO (Karpathy and Fei-Fei, 2017), as in previous studies.

# E Experimental Settings

## E.1 Main Experiment

Eight NVIDIA A100 80GB GPUs were used for training. The batch size was set to 8192, and 16 epochs of training were conducted using mixed precision training (Micikevicius et al., 2018) and gradient check pointing (Chen et al., 2016). The optimizer was AdamW (Loshchilov and Hutter, 2019), with a learning rate of 5e-4. The temperature parameter $\tau$ was fixed at 0.015625 (1/64). The average is the experimentally reported result.

The training on YFCC-CLIP used Inception-style random cropping (Szegedy et al., 2015), and the resolution was set to $224 \times 224$. The training on JWeb-5M changed only the lower limit of the scale of the crop range to 0.9 from the above settings. The lower limit of the scale of the crop range was changed to 0.9 when the model was trained on JWeb-5M. TrivialAugment Wide (Müller and Hutter, 2021) was used as the common augmentation method; image normalization CLIP (Radford et al., 2021) was used as well. The maximum input to the text encoder was 77 tokens. During testing, only resizing, cropping from the center, and normalization to a resolution of $224 \times 224$ were performed. The prompt text in the text encoder was not used for either training or testing. For DueT, the hyperparameter $m$ was set to 1536. We committed to $m = 1536$ early in our experiments as it consistently outperformed full fine-tuning. Meanwhile, when conducting multiple ablation tests simultaneously, it was revealed that for training on JWeb5M,

| Training set | Warm-up steps | #Epochs | Batch size |
|---|---|---|---|
| JWeb-50K | 1,000 | 64 | 2,048 |
| JWeb-100K | 1,000 | 64 | 2,048 |
| JWeb-300K | 2,000 | 64 | 2,048 |
| JWeb-500K | 2,000 | 64 | 4,096 |
| JWeb-1M | 1,000 | 32 | 4,096 |
| JWeb-3M | 2,000 | 32 | 4,096 |
| JWeb-5M | 2,000 | 16 | 8,192 |
| JWeb-10M | 2,000 | 16 | 8,192 |

Table 7: Training setup for each dataset

using $m = 768$ or even $m = 384$ proved to be sufficient, as shown in Table 2. Determining the optimal setting for $m$ remains a topic for future investigation.

### E.2 Other experimental settings

In an experiment to investigate the effect of different numbers of training data on model performance, the batch size, number of training epochs, and warm-up steps were set separately for each dataset. Table 7 shows the hyper-parameters set individually in the JWeb-k training.

## F Linear Probe evaluation

Here, we provide additional details about the linear probe experiments described in this paper, including the data sets and experimental setup used in the evaluation.

### F.1 Datasets

We used ten datasets from the well-studied evaluation suite described in (Kornblith et al., 2018) as well as five additional datasets in to assess the performance of models on a wide variety of distributions and tasks.

These datasets included MNIST, STL-10 (Coates et al., 2011), the German Traffic Sign Recognition Benchmark (GTSRB) dataset (Stallkamp et al., 2011), the Country211 (Radford et al., 2021), the Stanford Sentiment Treebanks(SST) dataset (Socher et al., 2013) .

The details on each dataset and the corresponding evaluation metrics are provided in Table 8.

### F.2 Experimental settings

We use image features taken from the penultimate layer of each model, ignoring the classification layer provided for uni-modal ViT. For the image encoders in DueT and FT, we used the features before the linear projection to the embedding space.

We trained a logistic regression classifier following to (Radford et al., 2021).

We train the logistic regression classifier using scikit-learn's L-BFGS implementation with up to 1,000 iterations and report the corresponding metric for each dataset. The strength of the L2 regularization $\lambda$ was determined using a hyperparameter sweep on the validation sets over the range between $10^{-6}$ and $10^{6}$ , with 96 logarithmically spaced steps. To save compute required for the sweeps, we perform a parametric binary search that starts with $\lambda = [10^{-6}, 10^{-4}, 10^{-2}, 1, 10^{2}, 10^{4}, 10^{6}]$ and iteratively halves the interval around the peak until it reaches a resolution of 8 steps per decade. The hyperparameter sweeps are performed on a validation split of each dataset. For the datasets that contain a validation split in addition to a test split, we use the provided validation set to perform the hyperparameter search, and for the datasets that do not provide a validation split or have not published labels for the test data, we split the training dataset to perform the hyperparameter search. For the final result, we combine the validation split back with the training split and report the performance on the unused split.

### F.3 Results

The individual linear probe scores are provided in Table 9. DueT achieved the best performance on 10 of the 15 datasets, i.e., it was included in the Cropper-Pearson 99.5% confidence interval around the top score for each dataset.

On many datasets, DueT and fine-tuning outperformed the uni-modal ViT, demonstrating the superiority of natural language supervision over traditional pre-training approaches based on image classification, as reported in (Radford et al., 2021). Furthermore, DueT performed better than fine-tuning; this result demonstrates the improved image recognition capability of. See Section 4.5 for a detailed discussion of the linear probe results.

## G Adaptive gating mechanism

To investigate better configurations of the gating mechanism, we implemented a token-level and sentence-level input-adaptive gating mechanism using a single-layer FFN. The gate value $\alpha^{l}$ of the expression 3 was changed to a one-layer FFN in order to evaluate the adaptive gate mechanism at the sentence and token levels.

| Dataset | Classes | Training size | Test size | Evaluation metric |
|---|---|---|---|---|
| Food-101 | 102 | 75,750 | 25,250 | accuracy |
| CIFAR-10 | 10 | 50,000 | 10,000 | accuracy |
| CIFAR-100 | 100 | 50,000 | 10,000 | accuracy |
| SUN397 | 397 | 19,850 | 19,850 | accuracy |
| Stanford Cars | 196 | 8,144 | 8,041 | accuracy |
| FGVC Aircrafts | 100 | 6,667 | 3,333 | mean-per-class |
| Describable Textures(DTD) | 47 | 3,760 | 1,880 | accuracy |
| Oxford-IIIT Pets | 37 | 3,680 | 3,669 | mean-per-class |
| Caltech-101 | 101 | 3,030 | 5,647 | mean-per-class |
| Flowers 102 | 102 | 2,040 | 6,149 | mean-per-class |
| MNIST | 10 | 60,000 | 10,000 | accuracy |
| STL-10 | 10 | 1000 | 8000 | accuracy |
| GTSRB | 43 | 26,640 | 12,630 | accuracy |
| Country211 | 211 | 43,200 | 21,100 | accuracy |
| SST2 | 2 | 67,349 | 1,821 | accuracy |

Table 8: Details on each dataset and the corresponding evaluation metrics

| Dataset | ViT | finetune | DueT |
|---|---|---|---|
| Food-101 | 79.05 | 86.04 | **88.64** |
| CIFAR-10 | 89.98 | 94.11 | **97.86** |
| CIFAR-100 | 73.36 | 81.36 | **88.88** |
| SUN397 | 69.40 | 70.38 | **73.72** |
| StanfordCars | 43.15 | **47.26** | 49.16 |
| DTD | 70.11 | 74.10 | 75.00 |
| MNIST | 96.52 | **98.2** | 97.92 |
| STL-10 | 98.42 | 97.96 | **99.47** |
| GTSRB | 56.12 | **80.55** | 77.21 |
| Country211 | 12.10 | **15.52** | **16.69** |
| SST2 | 54.48 | 54.75 | 55.35 |
| FGVCAircraft | 40.94 | 45.01 | 49.25 |
| OxfordPet | 86.95 | 86.44 | **90.0** |
| Flowers102 | 98.05 | 98.47 | 99.15 |
| Caltech101 | 90.93 | 91.02 | **94.78** |

Table 9: Linear probe performance of various pre-trained models on 15 datasets. Scores within the 99.5% Clopper-Pearson confidence interval of each dataset's top score are shown in bold.

Let $\mathrm{GAU}^l(\boldsymbol{H}^l)$ be defined as

$$\mathrm{GAU}^l(\boldsymbol{H}^l) = \mathrm{FFN}_\alpha^l(\boldsymbol{H}^l)\mathrm{FFN}^l(\mathrm{LN}(\boldsymbol{H}^l)) + (1 - \mathrm{FFN}_\alpha^l(\boldsymbol{H}^l))\boldsymbol{H}^l. \quad (5)$$

Correspondingly, $\mathrm{FFN}_{\alpha,\mathrm{sent}}^l(\boldsymbol{H}^l)$ and $\mathrm{FFN}_{\alpha,\mathrm{token}}^l(\boldsymbol{H}^l)$ are defined as follows:

$$\mathrm{FFN}_{\alpha,\mathrm{sent}}^l(\boldsymbol{H}^l) = \sigma(\boldsymbol{h}_{\mathrm{CLS}}^l{}^\top \boldsymbol{w}_{\mathrm{gate}}^l + b_{\mathrm{gate}}^l) \in \mathbb{R} \quad (6)$$

$$\mathrm{FFN}_{\alpha,\mathrm{token}}^l(\boldsymbol{H}^l) = \sigma(\boldsymbol{H}^l \boldsymbol{w}_{\mathrm{gate}}^l + b_{\mathrm{gate}}^l) \in \mathbb{R}^n, \quad (7)$$

where $\sigma$ is a sigmoid function. $\boldsymbol{w}_{\mathrm{gate}}^l \in \mathbb{R}^d$, $b_{\mathrm{gate}}^l \in \mathbb{R}$ is a trainable parameter and $n$ is the length of the sequences.

| | JWeb-test | | STAIR | | Flickr30k-JP | |
|---|---|---|---|---|---|---|
| Gate | I→T | T→I | I→T | T→I | I→T | T→I |
| N/A | 73.18 | 72.65 | 60.94 | 50.86 | 77.7 | 68.08 |
| FFN_token | **74.0** | **73.96** | 61.76 | 52.16 | 80.13 | **70.05** |
| FFN_sent | 73.62 | 73.46 | 61.38 | 52.36 | 80.13 | 69.27 |
| scalar | 73.17 | 73.21 | **61.89** | **52.53** | **80.4** | 69.27 |

Table 10: Performance of models trained with different gating mechanisms on JWeb-5M. $m = 1,536$.

Table 10 compares the results from models employing different gate mechanisms. The table shows that there is no clear performance improvement between the adaptive gating mechanism with the FFN and the gate factor.

## H LoRA tuning

This section describes the LoRA tuning implementation used in this paper and presents the unit-size evaluation.

### H.1 Implementation

We use $W_q$, $W_k$, $W_v$, and $W_o$ to denote the query/key/value/output projection matrices in the self-attention module of each transformer block. LoRA adds trainable pairs of rank decomposition matrices in parallel to existing weight matrices. We applied LoRA to $W_q$ and $W_v$ for all transformer blocks for both image and text encoders according to the (Hu et al., 2022). LoRA modules are added to all layers of the image and text encoders as in DueT. It also updates all parameters in the layer normalization of the pre-trained encoders during training.

| Method | #TP | STAIR I→T | STAIR T→I | Flickr30k-JP I→T | Flickr30k-JP T→I | IN-JP |
|---|---|---|---|---|---|---|
| $LoRA_8$ | 1.5 | 47.34 | 39.6 | 65.97 | 56.1 | 32.8 |
| $LoRA_{16}$ | 2.0 | 47.58 | 39.66 | 64.8 | **56.74** | **33.64** |
| $LoRA_{32}$ | 3.2 | 47.71 | **39.95** | 65.93 | 55.7 | **33.64** |
| $LoRA_{64}$ | 5.6 | **47.8** | 39.89 | **67.7** | 55.1 | 32.66 |
| $LoRA_{256}$ | 19.7 | 47.05 | 39.7 | 65.2 | 55.76 | 33.52 |
| $LoRA_{1024}$ | 76.4 | 47.01 | 39.17 | 66.53 | 55.88 | 32.82 |

Table 11: Performance of LoRA with varying adapter size on each test set.

## H.2 Unit-size evaluation on LoRA

The effect of the hyper-parameter $r$ was investigated for the LoRA used in this study. The results in Table 11 show that there is no significant difference in performance for different values of $r$. As discussed in section 4.3, while LoRA is effective when the amounts of training data is small and in few-shot tasks with very few additional parameters, its performance is far below that of DueT for the range of data used in this experiment. The results also show that increasing the number of additional parameters did not significantly change the performance, indicating that the method of adapter insertion and gate insertion, rather than the number of additional parameters, had a significant impact on performance.

## I Gates performance at each layer

The values of the gates in the case that only some of the adapters are inserted are plotted as blue and red lines in Figure 7. The gate values tend to be larger when adapters were inserted in the text encoder, and they were particularly large when the number of adapters was small. Considering the discussion in Section 4.4 on knowledge utilization and acquisition in all layers of the text encoder, it can be assumed that the total number of adapters embedding new knowledge decreased, and therefore, by increasing the gate value, the impact of the FFN is increased and new knowledge is embedded there. In the image encoder, when adapters were inserted only near the output layer, the gate values were not significantly different from those in the case when adapters are inserted in all layers. On the other hand, a different trend was observed when adapters were inserted only near the input layer. Specifically, when adapters were inserted only from the first to fourth layers (or from the first to eighth layers), the values of the gates in the third or fourth layers (or seventh or eighth layers) were larger compared to

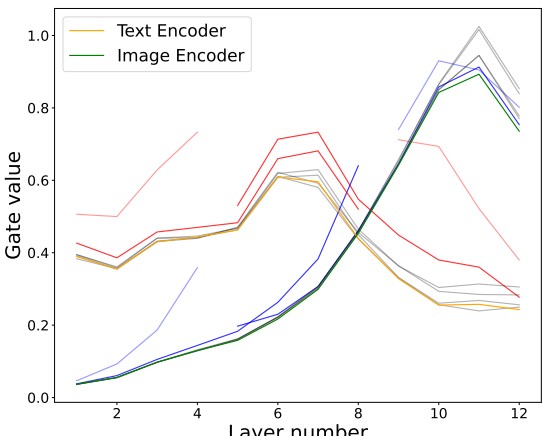

Figure 7: Gate values for each layer. Green: represents the gate values of the image encoder. Orange: represents the gate values of the text encoder. Blue: represents the gate values at the layer inserted within the image encoder. Red: represents the gate values at the inserted layer within the text encoder. Gray: represents the gate values of the image (or text) encoder with only some of the adapters are inserted in the text (or image) encoder.

the case when adapters were inserted into all layers. Given the above discussion, we conclude that this is also a result of the image encoder's tendency to utilize knowledge in the first half of the layer and to acquire knowledge in the latter half.

Lack of adapters around the output layer and lack of units to acquire new knowledge also explains the performance degradation when the adapters in the output layer of the image encoder are removed (Table 3).

## J Pre-trained models evaluation

We conducted a comparative analysis of different pre-trained models for model initialization. We trained the models on JWeb5M and used the same conditions as Section E for the rest of the experimental setup.

### J.1 Pre-trained models for image encoders

We compared five pre-trained models for initialization: AugReg* (Steiner et al., 2021) and CLIP Image Encoder (Radford et al., 2021), in addition to AugReg used in Section 4.3. AugReg* represents a model that was pre-trained on ImageNet-21k and then fine-tuned on ImageNet 2012. We trained the text encoder using BERT-base as initialization in all cases.

The results in Table 12 show that DueT outperformed fine-tuning in all cases, regardless of the pre-training model employed. These results sug-

| Model | Data. | Method | IN | I→T | T→I |
|-------|-------|--------|----|----|----|
| AugReg | IN21k | DueT | **42.85** | **71.15** | **60.9** |
| | | finetune | 37.23 | 69.84 | 57.67 |
| AugReg* | IN* | DueT | **41.86** | **71.2** | **60.79** |
| | | finetune | 35.65 | 69.08 | 57.69 |
| CLIP | WIT | DueT | **38.42** | **79.02** | **69.09** |
| | | finetune | 31.93 | 70.41 | 59.38 |

Table 12: Zero-shot transfer performance of models trained on JWeb5M with DueT and different image encoders that have the same ViT-B/16 architecture. I→T (T→I) shows the mean value of Recall@k (R@k; $k = 1, 5, 10$) on Flickr30k-JP and STAR.

| Model | Method | IN | I→T | T→I |
|-------|--------|----|----|----|
| BERT (ours) | DueT | **42.85** | **71.15** | **60.9** |
| | finetune | 37.23 | 69.84 | 57.67 |
| BERT* | DueT | **42.57** | **72.17** | **61.47** |
| | finetune | 36.36 | 67.71 | 57.94 |
| RoBERTa | DueT | **38.52** | 61.77 | **53.17** |
| | finetune | 34.82 | **62.26** | 52.66 |

Table 13: Performance of models trained on JWeb5M with different text encoders that have the same *base* architecture. We used AugReg-IN21k ViT-B/16 for the image encoder. We set $m = 1,536$. I→T (T→I) shows the mean value of Recall@k (R@k; $k = 1, 5, 10$) on Flickr30k-JP and STAR.

gest that DueT can selectively train new knowledge in GAU without forgetting existing knowledge, regardless of the knowledge embedded in the pre-training model.

### J.2 Pre-trained models for text encoders

We compared five pre-trained models for initialization: BERT*[7] and RoBERTa (Liu et al., 2019)[8] in addition to BERT (ours) used in Section 4.3. We trained the image encoder using ViT-B/16-AR-IN21k as initialization in all cases.

The results in Table 13 show that DueT outperforms finetune on all employed pre-training models except for the image to text search with RoBERTa. The results indicate that DueT is effective regardless of the architecture used in the text encoder or the knowledge embedded in the pre-training model.

---

[7]huggingface.co/cl-tohoku/bert-base-japanese-v3

[8]huggingface.co/rinna/japanese-roberta-base

## K Effect of varying the amount of training data

Table 14 shows detailed numerical values of the experimental results presented in Figure 4. These values show that DueT had the highest scores on all of the test sets when training was conducted with more than 1M data. On the other hand, LoRA outperformed DueT in STAIR, Flickr image-to-text retrieval, and ImageNet classification when the number of training data was relatively small, such as 50k or 100k, confirming that LoRA is effective when the number of training data is small.

## L Discussion on the Comparison with LilT

As mentioned in Section G, one of the points where DueT differs from LilT is the adoption of the gate mechanism, and the GAU validation results shown in Table 4 indicate that adapters that adopt the gate mechanism are effective in transition learning for the vision and language model. Preliminary experiments on the number and location of adapter insertions showed that the type adopted in this experiment, in which one GAU is inserted behind the FFN sub-layer, was the most effective.

Although the relationship between the insertion point and the gating mechanism, the number of training data and the effect of the training domain are not fully investigated, these experimental results suggest that DueT is expected to be a more effective method than LilT in the experimental setting we used.

| Train Data | Method | JWeb-test | | STAIR | | Flickr30k-JP | | IN-JP |
|---|---|---|---|---|---|---|---|---|
| | | I→T | T→I | I→T | T→I | I→T | T→I | |
| JWeb-50K | DueT | **26.93** | **26.63** | 23.74 | **19.61** | 42.7 | **35.66** | 13.01 |
| | LoRA$_8$ | 23.36 | 22.53 | **26.09** | 18.06 | **44.17** | 32.46 | **13.22** |
| | finetune | 25.61 | 25.0 | 21.63 | 17.46 | 40.2 | 31.65 | 12.13 |
| JWeb-100K | DueT | **33.55** | **33.59** | 30.25 | **24.49** | 47.6 | **39.27** | 16.55 |
| | LoRA$_8$ | 30.22 | 29.56 | **30.74** | 22.23 | **49.77** | 38.55 | **18.31** |
| | finetune | 31.85 | 31.36 | 27.43 | 21.34 | 43.57 | 37.88 | 17.13 |
| JWeb-300K | DueT | **44.98** | **44.69** | **39.6** | **31.41** | 54.8 | 47.27 | 23.16 |
| | LoRA$_8$ | 40.17 | 39.64 | 37.06 | 27.78 | **56.97** | **47.31** | **24.88** |
| | finetune | 43.74 | 43.73 | 34.07 | 28.37 | 51.0 | 43.5 | 22.85 |
| JWeb-500K | DueT | **50.79** | **50.87** | **44.85** | **35.69** | **62.27** | **54.12** | 25.24 |
| | LoRA$_8$ | 44.64 | 44.26 | 39.92 | 30.61 | 58.87 | 49.94 | 27.19 |
| | finetune | 48.74 | 48.63 | 39.76 | 33.06 | 56.63 | 49.03 | **27.68** |
| JWeb-1M | DueT | **57.77** | **57.77** | **48.89** | **41.21** | **68.37** | **59.34** | **30.76** |
| | LoRA$_8$ | 47.85 | 47.34 | 42.58 | 34.74 | 60.23 | 52.64 | 29.4 |
| | finetune | 54.2 | 54.36 | 44.5 | 34.99 | 62.8 | 52.37 | 22.61 |
| JWeb-3M | DueT | **68.58** | **68.87** | **58.66** | **48.88** | **80.27** | **67.97** | **39.09** |
| | LoRA$_8$ | 52.98 | 52.33 | 46.73 | 39.65 | 66.93 | 55.74 | 33.28 |
| | finetune | 66.17 | 66.42 | 56.19 | 44.72 | 73.17 | 60.81 | 31.49 |
| JWeb-5M | DueT | **73.17** | **73.21** | **61.89** | **52.53** | **80.4** | **69.27** | **42.85** |
| | LoRA$_8$ | 52.62 | 52.02 | 47.34 | 39.6 | 65.97 | 56.1 | 32.8 |
| | finetune | 72.12 | 72.95 | 60.98 | 50.29 | 78.7 | 65.04 | 37.23 |
| JWeb-10M | DueT | **78.76** | 78.51 | **65.02** | **55.77** | **84.53** | **73.04** | **46.81** |
| | LoRA$_8$ | 55.15 | 54.79 | 49.19 | 41.27 | 65.93 | 57.92 | 34.51 |
| | finetune | 78.42 | **79.11** | 64.81 | 54.84 | 80.87 | 68.84 | 40.86 |

Table 14: Performance of each task for different numbers of training data.