# OpenReview forum: "DueT: Image-Text Contrastive Transfer Learning with Dual-adapter Tuning"
_EMNLP/2023/Conference — EMNLP 2023 Main_

### Official Review · Reviewer_a7vW · 2023-08-04

**Soundness:** 4

**Excitement:**

4: Strong: This paper deepens the understanding of some phenomenon or lowers the barriers to an existing research direction.

**Paper Topic And Main Contributions:**

This paper presents a dual-adapter tuning (DueT). The authors add additional layers into both the image and text encoders and then train the added layers. The authors design a gating mechanism to the so called adapters.

The proposed method outperforms the baselines and LoRA-based adapter method in terms of accuracy and efficiency on image-text retrieval tasks. Linear probing results are encouraging.

**Questions For The Authors:**

- The authors should release their code and model to facilitate future research.


**Reasons To Accept:**

- The proposed method is simple and effective. It is technically valid and can enhance the image-text contrast learning framework.
- The authors describe their network design in detail, including the architecture of the proposed gated adapter unit.
- The authors considered an array of design choices for the baselines, and provided clear comparisons to demonstrate the effectiveness of DueT.
- Large-scale pre-training is conducted to validate the idea, including pre-training on public (YFCC) and private (JWeb) dataset, respectively.
- The paper is well-written and easy to understand.

**Reasons To Reject:**

I don't find serious concerns. The only minor concern is that adding GAUs would also increase training parameters, from 1% to 58%, depending on the hyper-parameter. But considering that all other layers are frozen, the increased num parameters are acceptable.

**Reproducibility:**

3: Could reproduce the results with some difficulty. The settings of parameters are underspecified or subjectively determined; the training/evaluation data are not widely available.

**Reviewer Confidence:**

4: Quite sure. I tried to check the important points carefully. It's unlikely, though conceivable, that I missed something that should affect my ratings.

---

> ### Author Rebuttal · Authors · 2023-08-29
>
> We appreciate you for highlighting the key aspects of our research. Regarding the issues raised about the augmentation of parameters, we concur with your positive feedback and maintain that the computational overhead is not a significant challenge in our model, particularly since the parameters outside the adapter layers are frozen.

---

### Official Review · Reviewer_pT3q · 2023-08-06

**Soundness:** 4

**Excitement:**

4: Strong: This paper deepens the understanding of some phenomenon or lowers the barriers to an existing research direction.

**Missing References:**

-

**Paper Topic And Main Contributions:**

This paper introduces DueT, an adapter-based method for aligning pre-trained vision and text encoders (ViT and BERT respectively) by inserting gated Adapter units within the frozen encoders and training them with a contrastive loss. By keeping the base Transformers frozen and only training the GAUs, the authors prevent catastrophic forgetting and preserve the unimodal capabilities of the language models. The authors pre-trained models on both English and Japanese image-text datasets, and evaluate on zero-shot image classification and image/text retrieval in both languages.

**Questions For The Authors:**

- Can more details be given about the hashing algorithm used?

- In Table 4, was alpha set to 0.2 or 0.02? I can't tell if the last row of this table is supposed to be the same as the base 0.2 setting (L364).

- Why was m=1536 selected? The performance does not seem particularly better than some of the lower values of m in Table 2. This choice of m should be explained in the Experimental setup, before the results.

**Reasons To Accept:**

- The paper does a good job of applying ideas from parameter-efficient fine-tuning tuning (PEFT) methods to the problem of image-text contrastive learning. The results demonstrate the effectiveness of the proposed solution, evidenced by the strong results on zero-shot retrieval and image classification.

- Experiments are conducted not only on the standard English benchmarks, but also on Japanese datasets. It is rare to come across a multimodal method that shows experiments on languages beyond English, so it's great to see a non-English language represented in the experiments.

- The methodology section is extremely well structured: 3.1 explains all the background well, 3.2 explains the GAU method well, and 3.3 motivates all the design choices very well, effectively explaining how different design choices correspond to different learning settings (L347-L359). Everything was well-written and extremely easy to follow.

- The various ablation studies in Section 4.4 are also very well-motivated, and experiments are designed accordingly -- specifically, how the size of the bottleneck layer affects the model's learning ability, the utility of GAUs in different layers, the analysis of gate values at different layers, the effect of gate initialization, and effect of amount of pre-training data. All of these are very interesting questions that have been addressed in the ablation study.

**Reasons To Reject:**

There are no real reasons to reject this work -- some additional details would be useful (mentioned in the Questions for Authors section), but these are not grounds for rejecting the work in my opinion.

**Reproducibility:**

4: Could mostly reproduce the results, but there may be some variation because of sample variance or minor variations in their interpretation of the protocol or method.

**Reviewer Confidence:**

4: Quite sure. I tried to check the important points carefully. It's unlikely, though conceivable, that I missed something that should affect my ratings.

**Typos Grammar Style And Presentation Improvements:**

-

---

> ### Author Rebuttal · Authors · 2023-08-29
>
> Thank you for emphasizing the contributions of our work and for your comments to further improve the paper.
> > Can more details be given about the hashing algorithm used?
> - We utilized the md5 algorithm to hash the bytestring of both the image and the text. However, other hashing algorithms, such as sha1, can also be used. We will include this information in the paper.
>
> > In Table 4, was alpha set to 0.2 or 0.02? I can't tell if the last row of this table is supposed to be the same as the base 0.2 setting (L364).
> - Thank you for pointing that out. Table 4 is correct, and L364 is an error. DueT initializes the gating value of alpha to 0.02.
>
> > Why was m=1536 selected? The performance does not seem particularly better than some of the lower values of m in Table 2. This choice of m should be explained in the Experimental setup, before the results.
> - As you pointed out, for training on JWeb5M, using m=768 or even 384 proved to be sufficient. We committed to m=1536 early in our experiments as it consistently outperformed full fine-tuning. Because we were concurrently conducting several ablation tests with this setting, we couldn't rerun the tests with the smaller m values this time. We will move the explanation regarding the choice of the m value to the experiment setup section

---

### Official Review · Reviewer_Tr3S · 2023-08-09

**Soundness:** 3

**Excitement:**

2: Mediocre: This paper makes marginal contributions (vs non-contemporaneous work), so I would rather not see it in the conference.

**Paper Topic And Main Contributions:**

The work proposes a novel method for transfer learning of vision and language models using adapter tuning with a gating mechanism. The method, called DueT, leverages the knowledge from pre-trained image and text encoders without modifying them and trains only the adapters. The work shows that DueT achieves better performance and parameter efficiency than existing methods for image and text retrieval in both English and Japanese domains. The work also provides insights into the optimal number of parameters, training data, and gates for transfer learning of vision and language models. The work contributes to the research on vision and language models and their applications in various services that require integrated visual and linguistic understanding.

**Reasons To Accept:**

1: The author suggested a method to use adapters with pre-trained image and text encoders. This reduced the number of parameters that needed to be trained. The adapters had gates that allowed effective transfer and connection of knowledge from the pre-trained encoders for each modality.

2: The author conducted experiments with various text and image encoders, adapter units, parameter initialization methods, and efficiency measures, following the proposed approach. The results demonstrate the effectiveness of the proposed method.

3: The analysis of solid ablation covered various aspects, such as parameter efficiency, the effectiveness of GAUs on different layers, the performance of gates, and so on

**Reasons To Reject:**

1: Using pre-trained image and text encoders for image-text contrastive learning is a common practice, lacking originality. There are many similar works in the literature.

2: Testing on a limited number of languages or training sets is not sufficient to support the claims

**Reproducibility:**

4: Could mostly reproduce the results, but there may be some variation because of sample variance or minor variations in their interpretation of the protocol or method.

**Reviewer Confidence:**

3: Pretty sure, but there's a chance I missed something. Although I have a good feel for this area in general, I did not carefully check the paper's details, e.g., the math, experimental design, or novelty.

---

> ### Author Rebuttal · Authors · 2023-08-29
>
> First and foremost, thank you for your review and the constructive feedback that aids in refining our manuscript.
>
> > 1: Using pre-trained image and text encoders for image-text contrastive learning is a common practice, lacking originality. There are many similar works in the literature.
>
> While our research utilizes contrastive learning and doesn't delve into introducing new learning theories or fundamental architectures from a technical novelty standpoint, we have incorporated new techniques, including the integration of GAUs, and carried out detailed ablation tests.
> Furthermore, as mentioned in Section 2.2 "Adaptation of CLIP to downstream tasks", while a majority of the research focuses on adapting pre-trained CLIP encoders for downstream tasks, our focus was on enhancing the image-text contrast learning framework using robust pre-trained uni-modal encoders.
> The significance of vision and language foundation models is on the rise. We are confident that our DueT method, which can obtain pre-trained models that achieve high performance with a low learning cost, provides a meaningful contribution to the field.
>
> > 2: Testing on a limited number of languages or training sets is not sufficient to support the claims
>
> As you rightly pointed out, recent research has been conducted on Multilingual-CLIP. Carlsson et al. (2022) trained a text encoder on machine-translated pairs of English and 67 other languages using teacher learning, thus enabling the encoder to accommodate non-English languages. Zhang et al. (2023) and Chen et al. (2023) adopted machine translation alignments and PEFT to train their text encoder, subsequently evaluating their approach on about 10 languages. Moreover, Khan & Fu (2023) demonstrated that by training with Multilingual-BERT on pairs of images and English text, it becomes feasible to attain low-resource adaptation for seven non-English languages.
>
> Our proposed method delivers a vision and language model that is particularly adept at handling the target language, in contrast to previous studies that aimed to create a single model for a multitude of languages.
> The following three points are the differences from previous studies and newly obtained findings.
>
> 1. Unlike previous studies, our method does not require a parallel corpus (a set of translated texts between two languages). The Duet model is trained simply using pairs of images and text in the target language.
>
> 2. The Japanese dataset used in our experiments is more expansive (reaching up to 10M) and encompasses a more diverse range of domains. The previous studies used images from public datasets such as Conceptual Captions, COCO, and Multi30k. Unlike these, our dataset introduces diverse visual concepts and encompasses a range of images, spanning shopping items, slides, PC screens, and Japanese characters, which remain absent from the pre-training domain of the ViT image encoder. We believe that the findings obtained from our experiments are new.
>
> 3. In experiments involving Japanese, we utilized a monolingual BERT pre-trained on Japanese text. In contrast, earlier studies often relied on multilingual models, be it Multilingual BERT or XLM-RoBERTa. Supplementary experiments, consistent with the specifications in Table 13, produced the following scores, indicating that the monolingual BERT outperforms the Multilingual-BERT:
>
> |          Model         | ImageNet  |  I -> T  |   T -> I  |
> |-----------------------|---------------|---------|----------|
> | Duet w/ BERT      |     42.85    |  71.15  |  60.90  |
> | Duet w/ M-BERT |     38.53    |  63.09  |  56.24  |
>
> Given the aforementioned considerations, while our experiments are limited to the linguistic scope of English and Japanese, we believe we have garnered new insights regarding "deep" adaptation to Japanese. Furthermore, our methodologies extend beyond language adaptation, encompassing domain adaptation and additional learning paradigms. We will integrate these discussions into both the main body and the appendix of our manuscript.
>
> References:
> * Fredrik Carlsson et al.: Cross-lingual and Multilingual CLIP. LREC 2022: 6848-6854
> * Zhen Zhang et al.: Parameter-Efficient Cross-lingual Transfer of Vision and Language Models via Translation-based Alignment. CoRR abs/2305.03510 (2023)
> * Guanhua Chen et al.: mCLIP: Multilingual CLIP via Cross-lingual Transfer. ACL (1) 2023: 13028-13043
> * Zaid Khan and Yun Fu: Contrastive Alignment of Vision to Language Through Parameter-Efficient Transfer Learning. ICLR 2023.

---

### Meta-Review · Area_Chair_LSWr · 2023-09-16

**Recommendation:** 5

**Metareview:**

This paper proposes an adapter-based method for aligning pre-trained vision and text encoders (ViT and BERT respectively) by inserting gated Adapter units within the frozen encoders and training them with a contrastive loss.

Two reviewers are positive about the paper, while one reviewer is negative about the paper. Overall, reviewers found that (1) the paper does a good job of applying ideas from parameter-efficient fine-tuning tuning methods to CLIP training; (2) paper is well written and structured; (3) experiments are comprehensive and ablation study is solid. The reviewer who gave a low score mainly complained about (1) lack of novelty; and (2) only test using the proposed method to train CLIP on two languages.

Generally, the AC is positive about the paper, and agrees that the negative comments did not hold, and that reviewer also did not participate in the discussion to back up his scores.

---

### Decision · Program_Chairs · 2023-10-07

**Decision:**

Accept-Main

**Comment:**

This paper proposes an adapter-based method for aligning pre-trained vision and text encoders (ViT and BERT respectively) by inserting gated Adapter units within the frozen encoders and training them with a contrastive loss.

Two reviewers are positive about the paper, while one reviewer is negative about the paper. Overall, reviewers found that (1) the paper does a good job of applying ideas from parameter-efficient fine-tuning tuning methods to CLIP training; (2) paper is well written and structured; (3) experiments are comprehensive and ablation study is solid. The reviewer who gave a low score mainly complained about (1) lack of novelty; and (2) only test using the proposed method to train CLIP on two languages.

Generally, the AC is positive about the paper, and agrees that the negative comments did not hold, and that reviewer also did not participate in the discussion to back up his scores.